# Development of a High-Fibre Multigrain Bar Technology with the Addition of Curly Kale

**DOI:** 10.3390/molecules26133939

**Published:** 2021-06-28

**Authors:** Hanna Kowalska, Jolanta Kowalska, Anna Ignaczak, Ewelina Masiarz, Ewa Domian, Sabina Galus, Agnieszka Ciurzyńska, Agnieszka Salamon, Agnieszka Zając, Agata Marzec

**Affiliations:** 1Department of Food Engineering and Process Management, Institute of Food Sciences, Warsaw University of Life Sciences, 159c Nowoursynowska St., 02-776 Warsaw, Poland; jolanta_kowalska@sggw.edu.pl (J.K.); ewa_domian@sggw.edu.pl (E.D.); sabina_galus@sggw.edu.pl (S.G.); agnieszka_ciurzynska@sggw.edu.pl (A.C.); agnieszkazajac230@gmail.com (A.Z.); agata_marzec@sggw.edu.pl (A.M.); 2Institute of Agriculture and Food Biotechnology—State Research Institute, 36 Rakowiecka St., 02-532 Warsaw, Poland; agnieszka.salamon@ibprs.pl

**Keywords:** snack, baking, carotenoids content, chlorophyll content, total polyphenols content, calorific value, sensory properties

## Abstract

The aim of this study was to find the effect of kale and dietary fibre (DF) on the physicochemical properties, nutritional value and sensory quality of multigrain bars. A recipe of multigrain bars was prepared with the addition of fresh kale (20% and 30%) and DF preparations (apple, blackcurrant, chokeberry and hibiscus). The bars were baked at 180 °C for 20 min. These snack bars, based on pumpkin seeds, sunflower seeds, flaxseed and wholegrain oatmeal, are a high-calorie product (302–367 kcal/100 g). However, the composition of the bars encourages consumption. In addition to the ability to quickly satisfy hunger, such bars are rich in many natural ingredients that are considered pro-health (high fibre content (9.1–11.6 g/100 g), protein (11.2–14.3 g/100 g), fat (17.0–21.1 g/100 g, including unsaturated fatty acids), carbohydrates (20.5–24.0 g/100 g), as well as vitamins, minerals and a large number of substances from the antioxidant group. The addition of kale caused a significant increase of water content, but reduction in the value of all texture parameters (TPA profiles) as well as calorific values. The content of polyphenols was strongly and positively correlated with the antioxidant activity (r = 0.92). In the bars with 30% addition of kale (422 mg GA/100 g d.m.), the content of polyphenols was significantly higher than based ones (334 mg GA/100 g d.m.). Bars with the addition of the DF were characterized by a higher antioxidant activity, and the content of carotenoids, chlorophyll A and B and polyphenols. High sensory quality was demonstrated for all (from 4.8 to 7.1 on a 10-point scale). The addition of fibre preparations was also related to technological aspects and allows to create attractive bars without additional chemicals.

## 1. Introduction

Consumers are increasingly turning to snack foods. This trend may be due to the fast pace of people’s lives, which results in a lack of time to prepare and eat traditional meals. In addition, they are paying attention to what they eat because they are more aware of the effects of food on their health. Cereal products, fruits and vegetables occupy an important place in the daily diet because they contain many valuable ingredients such complex carbohydrates, including dietary fibre (DF), vitamins, antioxidant compounds and minerals that can be considered pro-health. However, their consumption is still too low. Producers, to meet the requirements of consumers, introduce new, innovative products to the market such as cereal-based products with the addition of fruit or vegetables. Replacing traditional high-calorie snacks with products with a high content of bio-ingredients may have a beneficial effect on health. It can prevent many diseases, such as diabetes, arteriosclerosis, or high blood pressure.

Fresh and processed vegetables are a source of valuable nutrients. A greater degree of consumer attention should be focused on cruciferous vegetables, which are little appreciated. Kale is not a popular vegetable in the diet of consumers because of its taste, flavour and characteristic texture. Cruciferous vegetables are good sources of fibre, polyphenols, and glucosinolates. It is rich in biologically active substances with antioxidant, bactericidal and fungicidal properties [1]. This vegetable contains polyphenols, including flavonoids, which by inhibiting the activity of phosphodiesterase and cyclooxygenases can reduce platelet aggregation; therefore, it is recommended in atherosclerotic diseases. Glucosinolates may reduce the risk of cancer development [2]. Sulforaphane has the strongest anticancer properties. It shows an inhibitory effect on angiogenesis and the formation of metastases [3]. Kale is characterized by a high content of vitamins, such as in edible parts: C (120 mg/100 g), A (0.9 mg/100 g), B1 (0.1 mg/100 g), B2 (0.2 mg/100 g), B6 (1.6 mg/100 g), and E (1.7 mg/100 g), and also contains folic acid and niacin. In addition, it contains a large number of essential micro and macro elements, including, in edible parts: calcium (157 mg/100 g), potassium (530 mg/100 g) and iron (1.7 mg/100 g) [4,5]. Carotenoids, but also quercetin and camferol, are responsible for its strong antioxidant capacity [6]. The multitude of active compounds in kale translates into the health benefits of its consumption. Many studies have shown that frequent kale consumption reduces the incidence of cancer in various parts of the digestive tract, lung cancer and others. Son et al. [7], in view of the nutritional needs of patients with impaired renal function, attempted the production of kale with reduced potassium content without compromising the yield and quality. The potassium deficiency in kale was eaten up by an increase in total glucosinolate content, which is an indicator of the anticancer activity of cruciferous vegetables. Chlorophyll, contained in the raw material, has an antiseptic and immunizing effect [8,9]. The fresh leaves are suitable for direct consumption, and they can be added to salads or boiled as an ingredient of soups or fried. Michalak et al. [10] presented the possibility of using kale fermentation by autochthonous lactic acid bacteria in the creation of bioactive derivatives of phenolic compounds that may have anticancer properties. In recent years, it has also become a fashionable addition to juices or smoothies [11]. In the USA, powdered kale is used to make dietary supplements. On the local market, there are capsules with kale powder as an ingredient. 

Dietary fibre is a heterogeneous mixture of carbohydrate polymers found in plant raw materials. It refers to a large number of substances that exhibit a wide variety of physicochemical properties, with a general division into water-soluble and insoluble compounds. However, the way of processing, e.g., cereals, reduces its content in products. There is a need to make DF preparations that can be added to enrich various products. In addition, DF has important health and technological functions in food production. The raw materials for the production of DF preparations are industrial fruit and vegetable waste (apple pomace, blackcurrant pomace, waste from the processing of carrots, tomatoes) as well as bran, corn cobs, chaff and straw, and legumes (mainly soybeans and peas) [12]. The soluble fraction includes pectin, ß-glucans, gums, mucilage, and a wide range of indigestible oligosaccharides (including inulin). The insoluble fraction includes: lignin, cellulose, and hemicellulose. Each of the two types of fractions has different physiological effects. Soluble DF is less common in food than insoluble DF, but it has a significant impact on digestive and absorbent processes [12,13,14]. Fruit and vegetable DF has a much higher proportion of soluble DF, while cereal DF contains more insoluble cellulose and hemicellulose [15]. 

The addition of DF to bakery products, such as cereal snacks or multigrain snack bars, is justified both in terms of health and technology. Consuming products with DF helps to prevent many civilization diseases, such as: obesity, type 2 diabetes, ischemic heart disease, gallstone disease, constipation, and flatulence. It also reduces the risk of developing certain cancers. Its use in food production results from its ability to bind water, gel formation, emulating and stabilizing properties, and fat mimetic properties, therefore it is primarily a structure-forming component and filler and affects the sensory quality of food [12,16]. The European Food Safety Authority (EFSA) allows the nutrition claims “source of fibre” and “high fibre” on food packaging with a content of at least 3% (1.5 g/100 kcal) or at least 6% fibre, respectively (3 g/100 kcal) [17,18,19]. Food has an increased level of fibre when it is at least 25% higher than similar foods. Products that contain health claims for dietary fibre must also meet requirements for adequately low fat, including saturated fat and low cholesterol [20]. Products containing a large amount of DF can be classified as functional and/or health-promoting food. In the technology of producing granular bars, DF allows for a better, more compact combination of ingredients and obtaining the appropriate structure of the product, mainly due to water absorption and mechanical durability. When deciding to use a DF blend in the product, it should be taken into account that their effect on texture may vary depending on the product formulation.

The aim of the research was to evaluate the use of kale and fibre preparations as an added value to multigrain bars. The scope of the study was the effect of kale and dietary fibre (DF) on the physicochemical proper-ties, nutritional value and sensory quality of multigrain bars. 

## 2. Results

### 2.1. Development of the Recipe Composition and Production Technology of Multigrain Bars

At the stage of preliminary research, the recipe composition of the base bars was developed using baking. To improve the attractiveness of such a snack in terms of sensory and health, the recipe has been enriched with kale and DF preparations. Table 1 presents the chemical characteristics of fresh kale and multigrain bars with the addition of fresh and dried (microwave-blanched) kale. 

The raw material composition allowed to obtain a product with a DF content of 11.5%. Unrefined cereal products are characterized by a high content of DF, especially the insoluble fraction, similar to vegetables. Whole-grain oat flakes and flaxseed could play a significant role in the resulting DF content. The addition of kale did not have a significant effect on the differentiation of the bar composition in terms of DF content, but it caused a partial reduction in the caloric content of the bars (Table 1). The snacks obtained in this way can be a valuable source of both DF fractions, as well as fat (pumpkin and sunflower seeds) and protein. Due to their high calorific value, they can be a tasty and valuable snack that can replace one of the main meals.

Korus [21] analysed the composition of fresh vegetables. She showed a slightly higher content of proteins (about 4.3 g/100 g), emphasizing the beneficial composition of amino acids, including exogenous ones. Korus [5] showed that the pre-treatment of kale leaves reduced the content of minerals and vitamins by 26–52% (blanching) and 29–75% (cooking). The highest content of minerals, B vitamins and tocopherols was recorded in the frozen kale leaves after blanching. After 12 months of storage of frozen leaves, they contained 24–74% of macronutrients, 40–71% of micronutrients, 45–71% of vitamin B_1_, 27–47% of vitamin B_2_ and 69–85% of total tocopherols. In the study by Olsen et al. [22], green and red kale extracts have undergone a treatment including blanching, freezing and heat treatment by boiling in a bag. In both kale varieties, processing significantly decreased total phenolics, antioxidant capacity, and the content and distribution of flavonols, anthocyanins, hydroxycinnamic acids, glucosinolates, and vitamin C. Both extracts continued to inhibit colon cancer cell proliferation, but fresh kale extract had a much stronger effect. According to Korus [21], vegetables are low in fat, high in carbohydrates and DF and minerals, as well as vitamins and other important ingredients like antioxidants. Cruciferous vegetables deserve special attention. The author [21] showed that among all vegetables, this vegetable contains the most easily digestible calcium and protein and an exceptionally high amount of iron. Kale is also a significant source of DF, vitamins C and E (more than spinach and lettuce), provitamin A and antioxidants (more active than garlic, spinach, Brussels sprouts and broccoli).

Based on literature [5,21,22] data showing that kale processing reduces the content of many components, and because the content of individual ingredients was quite similar in both types of bars with fresh and dried kale (Table 1), the bars with 20 and 30% addition of fresh kale and DF preparations were used for a more detailed assessment of the physico-chemical properties. This was to make bars with lower calories and to facilitate the preparation of ingredients. Depending on the method of drying the kale, it could take up to several hours.

### 2.2. Water Activity and Its Content in Multigrain Bars

The water activity (Aw) in the bars was in the range of 0.857–0.953 (Table 2). This level of Aw classifies bars in the group of moist foods (in the range of 0.90–1.00) and with average Aw (in the range of 0.55–0.90), and thus in the food in which some microorganisms can develop (no microbiological stability). With the increase in the proportion of kale in the composition, the water activity increased. The base bars, i.e., bars without kale, were characterized by the lowest water activity (0.857). The type of added DF preparation had a significant effect on the water activity of the obtained bars (Table 2). Among the bars with 20% addition of kale, these with apple and chokeberry DF (0.943–0.944) were characterized by higher Aw, while bars with hibiscus DF (0.914) has significantly lower Aw.

The analysis of water content in the tested bars showed a significant effect of the type of added DF preparation and the amount of added kale on this parameter (Table 2). This varied widely, from approximately 17.3 to 41.1%. The water content in the control bars (without the addition of kale) differed significantly from the water content in the samples with the addition of kale. Increasing the proportion of kale from 20 to 30% resulted in a significant increase in the water content in the bars. 

### 2.3. The Effect of the Addition of Kale and Fibre Preparation on the Texture of the Multigrain Bars

#### 2.3.1. Compression Test

The texture of the bars was tested on the basis of the compression test and the work required was calculated. The control bars showed the highest compression work (604.3 mJ), about 2 times higher than the other bars (Table 2). A significant effect of increasing the amount of kale on the value of this indicator was demonstrated; for bars with 20 and 30% addition of kale, the work/deformation energy of the samples was about 383.5 and 271.4 mJ, respectively. Higher humidity decreased the hardness and the work needed for the deformation in the compression test was lower. The type of DF had a significant effect on the hardness of the bars. The bars with the addition of DF-Chokeberry were distinguished by significantly greater hardness (Fmax = 438.3 mJ), and P-Hibiscus by significant softness (268.6 mJ). This type of DF turned out to be less useful in the production of bars already at the stage of preparation before baking. 

#### 2.3.2. Texture Profile Analysis (TPA) Test

The addition of kale had a statistically significant effect on all parameters of the texture profile (Table 3), but no significant differences were observed in the amount of kale addition. The DF preparations had a significant effect on the parameters of the texture profile, only in the case of elasticity, no such effect was observed (Table 3).

The values of the hardness parameter for the tested bars ranged from 89 to 299 N. The control bars (298.2 N) had the highest hardness value, significantly lower values were achieved in bars with 20 (121.5 N) and 30% (94.6 N) kale addition. Therefore, it can be concluded that the kale bars were more brittle or soft than the base bars. However, the amount of kale added did not significantly affect the hardness of the bars. The results of this parameter were significantly influenced by the type of DF preparation used. The bars with the addition of apple and hibiscus DF had significantly lower hardness values than the bars with the addition of blackcurrant and chokeberry DF.

Elasticity discriminants of the tested bars changed, as did the hardness of bars with and without kale. The addition of kale decreased the value of this parameter. The base bars (without the addition of kale) had an elasticity of 0.54, the bars with 20% addition of kale 0.40, and the bar with 30% addition of 0.383. For bars with the addition of various DF preparations, the values of elasticity were at a similar level (0.40–0.48 N).

The addition of kale significantly influenced the cohesiveness. The bars without the addition (base) were characterized by higher cohesiveness than the bars with the addition of kale. The type of DF preparation also had a significant effect. The bars with the addition of apple DF and 20% kale had the lowest cohesiveness values (0.26), the bars with the addition of hibiscus powder (0.37), blackcurrant DF (0.43) and chokeberry DF (0.48) were characterized by higher values.

Guminess significantly depended on the addition of kale. The base bars (without addition) had higher values (145.0 N) than the bars with the addition of kale. The greater addition of kale did not cause any significant changes in the guminess of the bars. The effect of the various DF preparations was also significant. The bars with apple and hibiscus DF were characterized by low values of guminess (about 32 N), while the value of this parameter for bars with the addition of blackcurrant DF (75.7 N) and samples with the addition of chokeberry DF (88.8 N) was significantly higher.

The chewiness of the bars decreased significantly (to about 6 times) in the bars with the addition of kale compared to the control bars, from 77.0 to 12.9 N for bars with 20% additive and 14.4 N with 30% additive. Moreover, bars with apple DF (12.9 N) and hibiscus (13.8 N) were characterized by low chewiness values, significantly higher values were obtained for bars with blackcurrant and chokeberry DF addition.

### 2.4. The Effect of the Addition of Kale and Fibre Preparation on the Color of the Multigrain Bars

The addition of kale in the amount of 20 and 30% did not significantly affect the brightness of the colour of the bars (Figure 1a). On the other hand, the type of added DF preparation significantly influenced the brightness of the colour of bars. The addition of blackcurrant, chokeberry and hibiscus DF caused a significant darkening of the colour bars. The bars with the addition of apple DF showed up to about 16% higher values of the L* parameter. This dependence may be due to the colour of the DF preparations, apple DF is a light beige powder, while the remaining DF are dark red powders. Dark red discoloration of the preparations may be due to the presence of anthocyanin pigments. 

To evaluate the effect of the addition of kale and DF preparation on the colour changes of the bars, the absolute colour difference ΔE was calculated, and the colour of the control bars was set as a standard. The colour of the bars was clearly differentiated, because the absolute colour difference ranged from 6.3–13.7 (Figure 1a and Figure 2), from the colour of the bars with apple DF to those with blackcurrant and chokeberry DF.

The higher colour saturation is perceived by consumers as the more “alive”, while the lower the colour saturation, the more muffled it is, the closer to grey [23]. The values of the C* parameter in the bars tested ranged from 5 to 18% (Figure 1b). The base bars (approx. 17.5%) had higher values of the C* parameter, while the lower values of the bars with the addition of blackcurrant DF and chokeberry (approx. 5.5%). The addition of kale to the recipe significantly influenced the saturation of the colour of the bars, bars without the addition of kale had significantly higher values of the C* parameter than bars with the addition of 20 or 30% kale. The type of DF preparation used also significantly influenced the colour saturation. Bars with the addition of chokeberry and currant DF were characterized by the lowest values, significantly higher values were found for bars with the addition of hibiscus, while the most “vivid” colour was characterized by the bars with the addition of apple DF.

The colour hue of h bars was also analysed, which informs how much a given colour differs from white [23]. All tested bars had the h parameter in the range of 40–75°, which corresponds to the range of colours from red to yellow. Low h values were characteristic for bars with the addition of chokeberry DF (approx. 40°), higher bars with the addition of blackcurrant DF (approx. 50°) and bars with the addition of hibiscus (approx. 57°). The bars with the addition of apple DF had the highest colour shade values. The type of DF preparation used had a significant effect on the colour shade. The amount of kale addition used in the bar recipe was not significant. On the other hand, the addition of kale (20 and 30%) resulted in significantly higher h values than in bars without this additive.

### 2.5. The Effect of the Addition of Kale and Fibre on the Antioxidant Content of Multigrain Bars

The addition of kale significantly (Figure 3a) increased the antioxidant activity from 23.6 mM Trol/g d.m. for base bars without kale addition to 29.3 mM Trol/g d.m. for bars with 30% addition of kale. Bars with a 20% addition of kale and chokeberry DF (43.2 mM Trol/g d.m.) were characterized by high antioxidant activity, the value in these bars was nearly two times higher than in the based bars. It can be assumed that the DF preparation had a significant effect on this index. 

In the study by Korus [21], the antioxidant activity of kale was at the level of 14.7–23.7 μM Trol/g, so lower than in the bars tested. In terms of dry matter content, the results would be more similar. Moreover, the values of this indicator depend on many factors related to both the preparation of the product and the method of determination [24]. Due to the large number of different compounds influencing the antioxidant activity (soluble in water or organic solvents), the obtained results may depend on the method of preparation of the extract. The bars enriched with chokeberry DF had the highest value of polyphenol content (Figure 3b), i.e., about 552.5 mg GA/100 g d.m., significantly lower bars with the addition of hibiscus powder, about 457.5 mg GA/100 g d.m. For bars with the addition of blackcurrant DF and bars with the addition of apple DF, this value was in the range 377–396 mg GA/100 g d.m.

Korus [21] showed a high content of polyphenols in kale, at the level of 256–531 mg/100 g of fresh weight. Such a large range is influenced by the variety, growing conditions and maturity. This vegetable is therefore a very valuable source of these compounds, which are largely preserved in the bars.

Green vegetables are rich in chlorophyll and often contain carotenoids, the colour of which is not always discernible due to the predominant chlorophyll. In fresh kale tested by Korus [21], the total content of chlorophyll was 81–165 mg/100 g and carotenoids 16.8–34.2 mg/100 g. The control bars were characterized by a low content of chlorophyll (approx. 5.0 mg/100 g d.m.) and carotenoids (0.14 mg/100 g d.m.), and the addition of kale caused a 3–5 fold increase in chlorophyll content and 23–34 fold in carotenoids content (Figure 4a,b). This content increased with the increase in the percentage share of kale in the recipe. The DF preparations did not cause significant changes in the chlorophyll content in the bars. On the other hand, the content of carotenoids was significantly different, depending on the type of DF in the bar recipe (Figure 4b). The bars with the addition of blackcurrant DF contained significantly less (approx. 2.4 mg/100 g d.m.) carotenoids than the bars with the addition of chokeberry DF (approx. 4.1 mg/100 g d.m.).

### 2.6. The Effect of the Addition of Kale and Fibre on the Sensory Quality of the Multigrain Bars

The bars were positively assessed by potential consumers (Figure 5). Within the five sensory discriminants on a 10-point scale, apart from the overall quality of the chokeberry DF bars, all bars scored higher than 5.0, but not higher than 7.3 points. Most of the lower scores were given to bars with the addition of chokeberry DF (from 4.8 to 6.4 points). Most of the higher ratings were given to bars with P-Hibiscus, especially for texture and overall quality.

## 3. Discussion

Currently, the daily intake of DF by most consumers is too low. This is due to the high degree of processing of many products. DF is supplied mainly from cereals and cereal products, seeds of legumes and fruits and vegetables. These products differ not only in their DF content, but also in the type of DF compounds. They are found in vegetables and cereals are grouped into water-soluble (pectins, gums) and water-insoluble (cellulose, lignin, some of the hemicellulose) [25]. All bars contained fibre at the level above 9% (Table 1), which allows them to be classified as products with a high fibre content. Bars tested by Márquez-Villacorta and Vásquez [26] with a composition of 4.12% oat bran; 10.04% of pineapple peel powder and 17.18% of quinoa flakes contained more DF (13.28%) and protein (11.37%), and the overall acceptance score was slightly higher (7.47 points). Epidemiological studies suggest that regular consumption of fruits and vegetables containing both DF and natural antioxidant compounds may reduce the risk of many chronic diseases [27,28]. The current diet also focuses on the caloric content of food. Vatankhah et al. [29] investigated the suitability of stevioside, a natural low calorie sweetener, as a replacement for sucrose in Iranian sweet bread. They showed that the replacement of sucrose in the amount of 50%, the physical, chemical and sensory properties of the bread were similar to the base product, but the calorific value was reduced by 11%. Ibrahim et al. [30] assessed the possibilities of using date fruit in the bar recipe and replacing honey with date paste. With regard to the use of date paste up to 70%, bars with a share of 50% were characterized by the highest overall acceptability.

Due to their sensory value, wide availability and convenience, snacks are popular and frequently consumed products [31]. Consumers like snacks very much, but also pay increasingly attention to what they eat and are aware of the issue of healthy eating. The current market trends force the food industry to introduce such products that can be part of a healthy and balanced diet, but also tasty and encouraging consumption [32]. 

Depending on the composition, various methods are used for the production of snacks, which can generally be divided into the so-called “cold” and “hot”, respectively, without and with the use of increased temperature. The use of baking has benefits in terms of quality and product safety without the use of chemicals. During baking, starch gelatinization, browning reactions, changes in structure, surface properties and other mechanical behaviour of the bakery products occur [29], and the formation of their characteristic sensory properties. Based on the composition of the bars, they may be classified as snack bakery products. An additional advantage of choosing this method of producing bars was obtaining relatively soft products, such as bread. In order to increase the content of natural ingredients in multi-grain (wholegrain oatmeal, sunflower seeds, pumpkin seeds, flaxseed) bars, kale and DF were added to increase the high health-promoting potential of the bars, without chemicals. The taste of fresh kale does not encourage consumption, so an attempt was made to mask it. As a result, in the sensory evaluation, the addition of kale was less significant than the type of fibre. The addition of chokeberry to the fibre preparation was the least acceptable for most indications (4.8–5.2 points), but its flavour was distinguished (6.4 points). 

The water activity (Aw) in the bars was high (0.857–0.953), but when analysing bars with the addition of 20% kale, all Aw were below 0.95. No pathogenic microorganisms develop in such a product. Many bakery products are characterized by higher water activity [33]. The shelf life of the bars is short. To retain all the value of the snacks and extend their freshness, for example, modified atmosphere packaging should be used. Water is an essential ingredient in many foods. Affects a number of processes and reactions that can reflect the quality and stability of food during storage. Whether certain reactions will occur is primarily determined by the state of the water, which is characterized by its activity [34]. From the point of view of water activity, food can be divided into [35] wet with water activity in the range of 0.90–1.00, medium water content—water activity in the range of 0.55–0.90 and low water content—water activity in the range 0.00–0.55.

In general, a stable food is considered the one with a water activity in the range of 0.07–0.35. However, the development of microorganisms is almost completely limited already at the water activity below 0.60 [36].

With the increase in the share of kale in the recipe, the water content in the bars increased. This was due to the increase in the water content in the bar recipe, which was caused by the addition of kale containing about 85% water [6]. This form of kale addition was justified by the possibility of enriching the bars. However, from the technological point of view, the addition of kale in the amount of 20% was sufficient. The type of DF preparation significantly influenced the water content in the tested bars (Table 2). The water content in bars with the addition of currant and chokeberry DF was the lowest (about 27%), while the bars with the addition of hibiscus preparation (about 35%) had a significantly higher water content. This may indicate the different sorption properties of the DF preparations used. According to Miastowski et al. [36], water binding is one of the most important features of DF preparation. However, their large diversity in terms of the presence of DF compounds, depending on the source of origin, has a large impact on the degree of water binding in bars. This property also depends on the degree of micronization and the particle size composition. Therefore, the use of various preparations resulted in different values of water activity. This is advantageous in the manufacture of bars that should have the desired texture.

The water content has influenced the mechanical properties of bars. The correlation coefficient for the water content and compression force was about −0.92 (*p* = 0.0098), and for the compression work −0.94 (*p* = 0.0053). This proves a strong negative correlation between these properties and the water content.

The TPA test is used to test the texture of food based on indicators that reflect the consumer’s perception of the chewing experience [37]. For multigrain bars about 2 cm thick, softness is required and those up to about 1 cm thick can be crunchy. In the case of thin ones, especially those with increased carbohydrate content, one should aim to obtain a glassy (amorphous) state. The structure of the bars is influenced by the method of their production, especially the temperature value. In the research by Nikmaram et al. [38], the optimal conditions for the production of extrudates depended on the amount of sesame seeds added and the temperature of the process. The addition of kale and DF preparations had a statistically significant effect on the parameters of the texture profile. The higher content of sesame seed, incorporated into corn expanded extrudates, increased the hardness of the extrudates, possibly due to the content of fat, protein and fibre [38]. Similarly relationships Kowalczewski and Ivanišová [39] showed that the addition of fruit to the muffin recipe had a significant impact on the parameters of the texture profile. According to Wójtowicz and Baltyn [40], the hardness of snacks should be as low as possible, as it proves the fragility of these products. In the study by Kubiak and Dolik [41], the result for apples was 62.02 N, Wójtowicz and Balatyn [40], potato pancakes were characterized by a hardness in the range of 102–106 N, while in Heo et al. [42], muffins enriched with DF were characterized by a hardness of 412–491 N. In the research by Kubiak and Dolik [41], the bread was characterized by elasticity at the level of 0.94. The tested bars were characterized by almost two times lower elasticity. This may indicate their compact structure.

In a properly functioning organism, it is necessary to ensure a balance in so-called redox processes. If reactive oxygen species are not effectively quenched, it may lead to oxidative stress [39]. To prevent the formation and protect the body against reactive oxygen species, one should eat food rich in antioxidant compounds [43]. Polyphenols are compounds synthesized by plants. Several thousand compounds belong to the group of phenol compounds, but they all have one thing in common, which is the antioxidant properties. The content of polyphenols was strongly and positively correlated with the antioxidant activity (r = 0.92). The use of kale addition caused changes in the content of polyphenolic compounds (Figure 3b). In the case of 30% addition of kale (422 mg GA/100 g d.m.), the content of polyphenols was significantly higher than for bars without addition (334.6 mg GA/100 g d.m.). The amount of polyphenolic compounds largely depended on the DF preparation used. The bars enriched with chokeberry DF had the highest value, i.e., about 552 mg GA/100 g d.m. In the study by Nawirska et al. [44], chokeberry pomace was also characterized by the highest polyphenol content among the tested fruits, and significantly lower values were obtained for blackcurrant. Biegańska-Marecik et al. [45] showed that kale has one of the highest values of antioxidant activity. The addition of frozen and freeze-dried kale on beverages based on apple juice resulted in a two- and three-fold increase in antioxidant activity, respectively. Murugesan et al. [1] showed that the antioxidant capacity of kale leaf ethanol extract was 62.9% (DPPH*), and GC-MS chromatographic analysis included profiles of more than 17 major phytochemicals in the extract. Additionally, Satheesh et al. [46] reported that there has been a growing trend in recent times to include more green leafy vegetables in the human diet, and kale has great potential for use in a variety of food and nutritional applications. Kale has been shown to have the nutritional and anti-nutritional components of kale, with research showing its multiple health benefits.

The anthocyanin pigments present in the bars had a positive effect on the antioxidant activity, but they could cause colour changes of the product. Colour is a parameter that has a large impact on the perception of food by consumers, as it can reflect the quality of food products. According to Kowalczewski and Ivanišová [39], these changes may not be accepted by consumers for some products. Moreover, in the case of bars, it can be noticed that the addition of chokeberry DF preparation, which had a positive effect on the content of polyphenols, translated into high oxidative activity and caused colour changes (Figure 1a,b). In addition, the bars were darker because the L* values were lower (Figure 1a). However, no significant differences were observed in consumer assessments regarding the colour of the bars (Figure 5).

Carotenoids and chlorophylls are plant pigments that give colour to vegetables and fruits, they are located in chloroplasts. Carotenoids are responsible for the red, orange and yellow colours. They are considered one of the strongest antioxidants and are credited with the ability to extinguish free radicals. They are also precursors of vitamin A. Chlorophylls are credited with bacteriostatic and anti-inflammatory properties, supporting the removal of carcinogenic toxins and with antioxidant properties. They give plants and products a characteristic green colour [47].

According to Karwowska et al. [47], fresh kale is characterized by the content of chlorophyll A and B at the level of 904.5 mg/100 g d.m. Kale added to the recipe increased the content of chlorophyll in the bars. This content increased with the increase in the percentage share of kale in the recipe. The addition of kale in the amount of 20 and 30% resulted in a 3.6 and 5.3-fold increase in their chlorophyll content, respectively, in comparison with the control samples (approx. 5.0 mg/100 g d.m.). The DF preparations of blackcurrant, chokeberry and hibiscus did not cause significant changes in the chlorophyll content in the bars. However, in the bars with the addition of chokeberry and hibiscus DF, the content of chlorophyll was lower by 10–14% than for those with chokeberry DF.

The addition of kale enriched bars with carotenoids. As in the case of chlorophyll, the addition of kale caused a greater increase in the carotenoid content. The bars with 20% additive (3.2 mg/100 g d.m.) contained about 23 times more dye than the base bars (0.136 mg/100 g d.m.). Bars with more kale contained the most carotenoids (4.7 mg/100 g d.m.). According to Karwowska et al. [47], it contains carotenoids in the amount of about 175 mg/100 g d.m. The effect of the DF preparation used on the content of dyes was also observed. The bars with the addition of blackcurrent DF contained significantly less carotenoids than the bars with the addition of chokeberry DF. Aronia contains 140–230 mg of carotenoids per 100 g of d.m., while blackcurrant only 20–40 mg/100 g of d.m. [48]. This translated into the final content of these dyes in the baked bars.

## 4. Materials and Methods

### 4.1. Material

The materials for the research were multi-grain bars with the addition of fresh kale prepared according to the established recipe (Table 4). The type of DF added and the percentage of fresh kale (*Brassica oleracea* L. var. *acephala*) added was variable in the recipe. The raw materials were purchased in a large-area store, while the DF preparations were obtained directly from the producer (GreenField, Poland). 

### 4.2. Experimental Procedure

#### 4.2.1. Preparation of BARS

The dry ingredients were ground in a grinder (Bosch MKM6000) for 20 s. Ground flaxseed was poured over with hot water to gel. The kale was ground in a Thermomix TM 31 device (Vorwerk Ltd., Wroclaw, Poland) for 10 s, speed of rotation—level 7. Then, all ingredients were combined and mixed for about 2 min. After receiving the mass, it was placed in rectangular form with dimensions of 100 × 40 × 20 mm. The formed bars were baked or dried in three different variants.

#### 4.2.2. Baking

Baking was carried out in an electric Piccolo oven (Winkler Wachtel Ltd., Wroclaw, Poland) for 25 min. The temperature of the lower and upper chamber of the furnace was 180 °C.

### 4.3. Analytical Methods

#### 4.3.1. Determination of Dry Matter Content

The dry matter content of the bars was determined by drying in a laboratory dryer (WAMED SUP-65 WG, Warsaw, Poland) at 130 °C for 1 h. The vessels with/without samples were weighed on an analytical balance (ME54E/M, Metler, Warsaw, Poland) with an accuracy of 0.001 g. The measurement was performed in duplicate.

#### 4.3.2. Determination of Water Activity

Water activity was determined with an AQUALAB CX-2 device (Decagon Devices Inc. Pullman, WA, USA). Measurements were carried out at the temperature of 23 ± 1 °C. The measurement was performed in duplicate, the final result was the mean of the measurements.

#### 4.3.3. Colour Parameters

The colour of the bars was measured with the Konica Minolta CR-300 colorimeter (standard observer CIE 2°, illuminat D65, measuring gap 8 mm) in the CIE Lab system. The measurement was performed in 5 replications. The mean of the measurements was taken as the result.

#### 4.3.4. Examination of Bars Structure

##### Compression Test

The mechanical properties were tested in a TA-HD plus texturometer (Stable Micro Systems, Godalming, UK). The compression test was performed with a 75 mm diameter head. Bars with dimensions of 25 × 40 × 20 mm were used for the measurement. The head speed was 1 mm/s. The samples were compressed to 50% of their height. The measurement was performed in 10 replications. The compression test was performed for the bars 4 h after the end of drying or baking. On the basis of the test, the compression work calculated as the product of the half of the area under the deformation curve and the head travel speed were determined.

##### Texture Profile Analysis (TPA) Test

The texture profile test was performed with a TA-HD plus texturometer (Stable Micro Systems, Godalming, UK). The measurement was performed 4 h after the end of drying or baking. Bars with dimensions of 25 × 40 × 20 mm were used for the measurement. The tested samples were compressed twice to about 50% of the original height. The head speed was 1 mm/s. The measurement was performed in 10 replications. On the basis of the test, the mechanical determinants of texture, such as hardness, elasticity, cohesiveness, gumminess, and chewiness, were determined as follows:■Hardness—the maximum value of the force used during the test [N];■Elasticity—the ratio of the compression time in the second cycle to the compression time in the first test cycle;■Cohesiveness (compressibility)—the ratio of the compression work in the second cycle to the compression work in the first cycle;■Gumminess—the product of hardness and cohesiveness [N];■Chewiness—product of gumminess and elasticity [N].

#### 4.3.5. Chemical Determinations

Chemical determinations were carried out in an accredited laboratory at the Institute of Agriculture and Food Biotechnology—State Research Institute in Warsaw, Poland. All determinations were performed at least in duplicate.

##### Nutritional Value

Determination of the nutritional value, i.e., protein, fat, ash, DF and carbohydrates, was carried out in the accredited laboratory of the Institute of Agricultural and Food Biotechnology—National Research Institute in Warsaw Poland. All determinations were performed at least twice.

##### Determination of Protein Content

Total nitrogen content was determined by the reference titration method (Kjeldahl) and converted into total protein content, taking into account the nitrogen to protein conversion factor 6.25 according to PN-EN ISO 20483: 2014 standard.

The principle of the method consists in converting organic nitrogen compounds contained in a dry sample of ammonium sulphate with concentrated sulfuric acid in the presence of a catalyst, basifying the solution, distilling and titrating ammonia bound in boric acid with the addition of indicators with sulfuric acid.

##### Determination of Fat Content

The fat content was determined in accordance with the PN-A-79011-4: 1998 standard. The principle of the method is based on the extraction of fat from a dry sample under predefined conditions using petroleum ether by means of a Soxhlet apparatus, and then weighing the residue of the sample after complete evaporation of the solvent.

##### Determination of Ash Content

The ash content was determined by the gravimetric method after the samples were incinerated according to the PN-EN ISO 2171: 2010 standard. The principle of the method is based on incineration of the dry sample (pre-dried) at the temperature of 900 °C and determination of the inorganic residues after ashing by weight.

##### Determination of Total Dietary Fibre Content

The total dietary DF content, including the soluble and insoluble fractions was determined by the gravimetric method after prior enzymatic hydrolysis of the samples using the Megazyme Total Dietary Fibre Kit (Bray, Bray Business Park, Co. Wicklow, A98 YV29, Ireland).

##### Calculation of Carbohydrates Content, Including Sugars

The carbohydrate content (CC) in g/100 g d.m. was calculated from the formula:CC = 100 − (H + A + F + P + DF)(1)
where:
H—humidity of the sample, [g/100 g d.m];A—ash content of the sample, [g/100 g d.m];F—fat content of the sample, [g/100 g d.m];P—protein content of the sample, [g/100 g d.m];DF—dietary fibre content of the sample, [g/100 g d.m].

Determination of the content of individual sugars, including: fructose, glucose, disaccharides (sum of sucrose and maltose) was performed using high performance liquid chromatography (HPLC) with refractometric detection of sugars contained in the aqueous solution obtained from the sample. The result of the sugar content was given as the sum of individual sugars [g/100 g d.m.].

##### Calculation of Energy Value

The energy value (EV) of the product (bars) was calculated on the basis of the energy content of protein (1 g = 4 kcal or 17 kJ), carbohydrates (1 g = 4 kcal or 17 kJ), fat (1 g = 9 kcal or 37 kJ) and dietary fibre (1 g = 2 kcal or 8 kJ) contained in them. The energy value in kcal/100 g and kJ/100 g of product was calculated [49]:EV [kcal /100 g] = (P + CC) ⋅ 4 + F ⋅ 9 + DF ⋅ 2(2)
EV [kJ/100 g] = (P + CC) ⋅ 7 + F ⋅ 37 + DF ⋅ 8(3)
where:
P—protein content of the sample [g/100 g];CC—carbohydrates content of the sample [g/100 g];F—fat content of the sample [g/100 g];DF—dietary fibre content of the sample [g/100 g].

##### Determination of DPPH Radical Scavenging Activity

The antioxidant activity (AA) was determined using the spectrophotometric method with the DPPH radical based on the method of Urbańska at el. [50] and Wong at el. [51]. For the preparation of samples, 2.4 mL of DPPH methanolic radical solution (60 μM) was used and 100 μL of acetone extract of the samples (the extract was prepared in the same way as for the determination of carotenoids/chlorophylls) was added. The samples were mixed and incubated at room temperature for 30 min in the dark. After this time, the absorbance was measured at the wavelength λ = 515 nm against the blank. The acetone solution and DPPH solution were collected for the control sample. The blank was a sample containing of methanol and of 80% acetone.

The antioxidant activity (quenching/scavenging capacity) of the DPPH radical (% inhibition) was calculated:(4)$$\%inhibition=\frac{A_{0}-A_{1}}{A_{0}}\cdot 100$$
where:
*A*_1_—absorbance of the DPPH radical with acetone extract from the sample;*A*_0_—absorbance of the DPPH radical with acetone (control sample).

When the calculated inhibition was greater than the 95% value, the sample was diluted with 80% (*v*/*v*) acetone solution so that the absorbance value was linear over the range of the analysed concentrations.

The antioxidant activity (AA) based on the DPPH free radical scavenging ability of the extract was expressed as mM Trolox per 1 g of dry matter (d.m.) of the sample. 

##### Determination of Total Polyphenol Content by the Folin–Ciocaletau Method

The total polyphenol content was determined by spectrophotometric method with the use of the Folin–Ciocaleteu reagent, which consisted of a coloured reaction of polyphenolic compounds with this reagent [50]. To the test tube was added 15% sodium carbonate (0.5 mL), distilled water (8.9 mL), acetone extract of the sample (0.5 mL; the extract was obtained in the same way as for the determination of carotenoids/chlorophylls, chapter 3.3.6), and 100 μL of Folin–Ciocalteu reagent. The sample was then mixed and incubated for 45 min in the dark (room temperature). After this time, the absorbance was measured at the wavelength λ = 765 nm against the blank. When the measured absorbance of the sample was greater than 0.650 value, the sample was diluted with 80% (*v*/*v*) acetone solution. The determination was performed in duplicate. The content of total polyphenols was expressed as mg of gallic acid (GA) per 100 g dry matter (d.m.) of the sample. 

##### Determination of Carotenoids and Chlorophyll A and B Content

Determination of carotenoids and chlorophyll content in the bar samples was performed using the BECKMAN DU-530 spectrophotometer (Beckman, UK). The samples were milled with a Sencor grinder to obtain the extract. An 80% (*v*/*v*) acetone solution (25 mL) was added to the weighed sample (about 1.0 g). The samples were homogenized for 30 s at a speed of 13,500 rpm in an ULTRA-TURRAX T25 basic homogenizer (IKA-WERKE, Germany). Then, the obtained homogenate was centrifuged in a laboratory centrifuge MPW 375 (MPW-Med-Instruments, Poland) for 3 min at a speed of 10,000 rpm. The measurements were made for chlorophyll A at wavelengths λ = 663 nm, for chlorophyll B λ = 647 nm, and at λ = 470 nm for carotenoids with the blank, which was an 80% (*v*/*v*) acetone solution. When the measured absorbance of the sample was greater than 0.900 in value, the sample was diluted with an 80% (*v*/*v*) acetone solution. The determination was performed in duplicate. The content of carotenoid pigments, chlorophyll A and B in the acetone extract was calculated from the equations [52]:(5)$$C_{C}=\frac{1000\;\cdot \;A_{470}-1.82\;\cdot C_{A}-85.02\;\cdot \;C_{B}\;}{198}$$
(6)$$C_{A}=12.25\;\cdot \;A_{663}-2.79\;\cdot \;A_{647}$$
(7)$$C_{B}=21.50\;\cdot \;A_{647}-5.10\cdot \;A_{663}$$
(8)$$C_{A+B}=7.15\cdot A_{663}+18.71\cdot A_{467}$$
where:*C_C_*—content of carotenoids in acetone extract [μg/mL];*C_A_*—content of chlorophyll A in acetone extract [μg/mL];*C_B_*—content of chlorophyll B in the acetone extract [μg/mL];*C_A+B_*—content of chlorophyll (total A + B) in the acetone extract [μg/mL];*A*_663_—absorbance of acetone extract measured at wavelength λ = 663 nm;*A*_647_—absorbance of acetone extract measured at a wavelength of λ = 647 nm;*A*_470_—absorbance of acetone extract measured at wavelength λ = 470 nm.

The content of chlorophyll or carotenoid dyes in the sample was calculated in mg per 100 g dry matter (d.m.). All determinations were performed at least in duplicate.

##### Sensory Evaluation

The sensory evaluation was performed by a team of 30 unqualified people, aged 18 to 45, using a 10-point scale. The evaluators were instructed on how to evaluate the selected discriminants such as taste, colour, smell, texture, and overall desirability (Table 5).

### 4.4. Statistical Analysis

The statistical analysis of the obtained results was performed with the use of Microsoft Excel and STATISTICA 13 PL programs. To determine the effect of the amount of curly kale and the addition of DF on the selected indicators, a one- or two-factor analysis of variance and Tukey’s HSD test were performed to determine homogeneous groups (post hoc test). Pearson’s correlation was also performed to investigate the relationship between the selected indicators.

## 5. Conclusions

The addition of kale and DF preparations had a beneficial effect on the physico-chemical, sensory and pro-healthy properties of snacks. In the production of bars, DF also played a technological role, enabling the appropriate consistency of the mix before baking and the texture of the final products. Multigrain raw materials are characterized by a high content of DF. Multigrain bars with the addition of kale and DF preparations can be a valuable source of both DF fractions, antioxidant compounds, as well as fat, protein, vitamins and minerals. They can be an offer of snacks for people struggling with health problems, as well as for healthy people who are looking for tasty and valuable products.

## Figures and Tables

**Figure 1 molecules-26-03939-f001:**
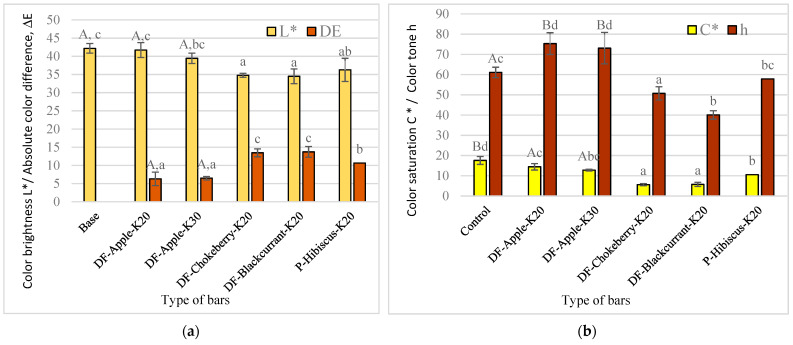
The influence of kale and fibre preparation addition on the colour parameters of multigrain bars: (**a**) Colour brightness L* and absolute colour difference ΔE, (**b**) colour saturation C* and colour tone h. Designation: a, b, c, d—homogeneous groups, the influence of: fibre preparations and A, B—kale at α = 0.05. The bar codes are explained in Table 4.

**Figure 2 molecules-26-03939-f002:**
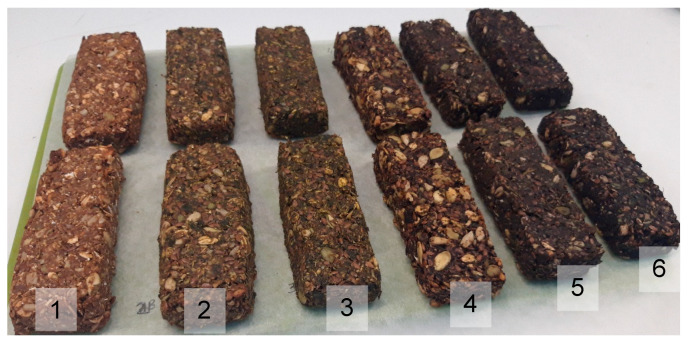
Pictures of baked bars, respectively: 1—base (Control), 2—with apple fibre and 20% kale, 3—with apple fibre and 30% kale, 4—with hibiscus fibre and 20% kale, 5—with fibre from blackcurrant and 20% kale, 6—with chokeberry fibre and 20% kale.

**Figure 3 molecules-26-03939-f003:**
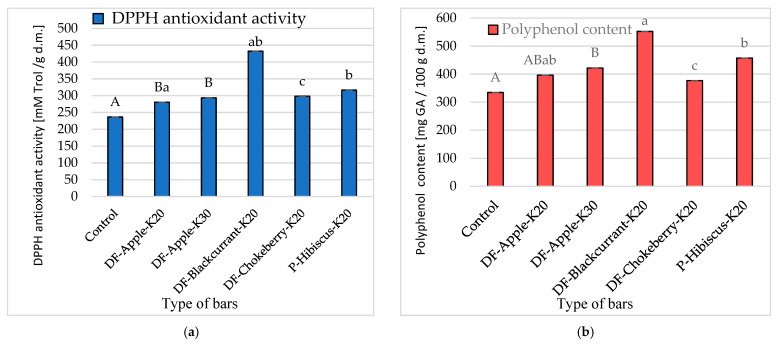
The influence of the kale (factor I) and fibre preparations (factor II) addition in multigrain bars on the: (**a**)—DPPH antioxidant activity [mM Trol/ g d.m.] and (**b**)—polyphenol content [mg GA/100 g d.m.]. Designations: A, B—homogeneous groups (factor I), and a, b, c—(factor II) at α = 0.05. The bar codes are explained in Table 4.

**Figure 4 molecules-26-03939-f004:**
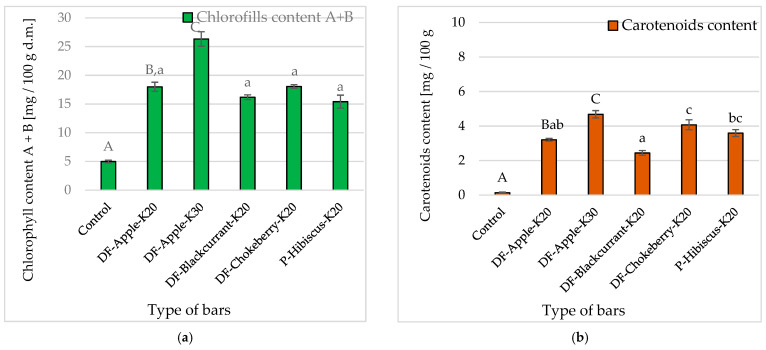
The influence of the kale (factor I) and fibre preparations (factor II) addition in multigrain bars on the: (**a**)—chlorophyll content A + B [mg/100 g d.m.] and (**b**)—carotenoids content [mg/100 g d.m.]. Designations: A, B, C—homogeneous groups (factor I), and a, b, c—(factor II) at α = 0.05. The bar codes are explained in Table 4.

**Figure 5 molecules-26-03939-f005:**
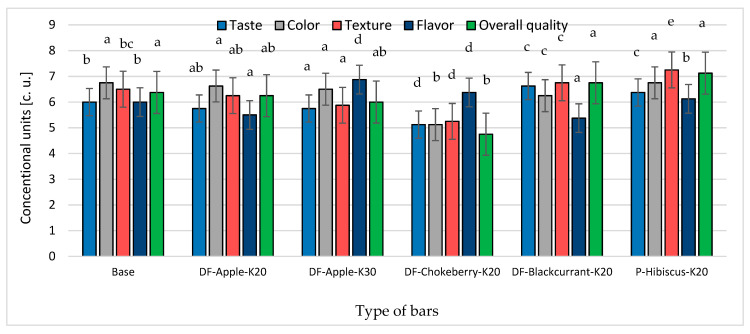
The effect of kale and fibre preparation or powder of dried hibiscus flower addition on sensory properties of multigrain bars. Designations: a, b, c, d, e—homogeneous groups (the influence of type of bars) at α = 0.05. The bar codes are explained in Table 4.

**Table 1 molecules-26-03939-t001:** Chemical characteristics of kale and multigrain bars with the addition of fresh and dried kale (microwave-blanched).

Chemical Characteristics	Fresh Kale	Base Bar − Baking	Bar + Kale 30%	Bar + Dried Kale 10%
Humidity, g/100 g	87.1	27	39.2	30.5
Total protein (N × 6.25), g/100 g	3.3	14.3	12.2	13.9
Total protein (N × 6.25), g/100 g d.m.	25.9	19.6	20.1	20
Fat (no hydrolysis), g/100 g	0.4	21.2	17	19.2
Fat (no hydrolysis), g/100 g d.m.	3.2	29	28	27.7
Total ash, g/100 g	1.1	1.9	1.9	2.3
Total ash, g/100 g d.m.	8.3	2.6	3.2	3.3
Total dietary fibre, g/100 g	5.1	11.6	9.1	11.4
Total dietary fibre, g/100 g d.m.	39.5	15.8	15	16.4
including: soluble fibre fraction, g/100 g	0.3	2.5	2	2.2
soluble fibre fraction, g/100 g d.m.	2.4	3.4	3.3	3.1
insoluble fibre fraction, g/100 g	4.8	9.1	7.1	9.2
insoluble fibre fraction, g/100 g d.m.	37	12.4	11.7	13.3
Total carbohydrates, g/100 g	3	24	20.5	22.6
Total carbohydrates, g/100 g d.m.	23.3	32.9	33.8	32.5
including: sugars, g/100 g	2.6	12.9	10.6	12.5
sugars, g/100 g d.m.	19.9	17.7	17.5	18
Energy value, kcal/100 g	39	367	302	342
Energy value, kJ/100 g	164	1528	1259	1424

**Table 2 molecules-26-03939-t002:** The effect of kale and fibre preparation on water content, water activity (Aw), and compression force (Fmax) and work (W) in multigrain bars.

Type of Bars	Water Activity [-]	Water Content [%]	Compression Work [mJ]
Control (based)	0.857 ± 0.000 ^A^	17.34 ± 1.13 ^A^	604.3 ± 33.6
DF-apple-K20	0.944 ± 0.001 ^c^^B^	29.86 ± 0.87 ^ab^^B^	383.5 ± 64.3
DF-apple-K30	0.953 ± 0.003 ^C^	41.12 ± 2.63 ^C^	271.4 ± 21.2
DF-blackcurrant-K20	0.931 ± 0.001 ^b^	27.48 ± 2.27 ^a^	360.3 ± 45.2
DF-chokeberry-K20	0.943 ± 0.000 ^c^	27.54 ± 2.57 ^a^	438.3 ± 41.4
P-hibiscus-K20	0.916 ± 0.001 ^a^	35.39 ± 0.23 ^b^	268.6 ±95.8
**One-way analysis of variance (ANOVA)**			
Factors	*p*-value		
Kale addition (A,B,C)	0.0001 *	0.0020 *	0.0000 *
Type of fibre (a,b,c)	0.0001 *	0.0310 *	0.0138 *

*—means significant difference at a confidence level of 0.05; a, b, c and A, B, C—homogeneous groups, the same letters mean no statistically significant differences between the analysed values of indicators; the codes are described in Table 4.

**Table 3 molecules-26-03939-t003:** The effect of kale and fibre preparation on the texture profile of multigrain bars. The bar codes are explained in Table 4.

Type of Bars	Hardness [N]	Elasticity [-]	Cohesiveness [-]	Guminess [N]	Chewing [N]
Control	298.2 ± 25.4 ^B^	0.54 ± 0.03 ^B^	0.48 ±0.064 ^B^	145.0 ± 29.0 ^B^	77.0 ± 15.7 ^B^
DF-Apple-K20	121.5 ± 16.0 ^aA^	0.40 ± 0.03 ^A^	0.26 ± 0.021 ^aA^	32.2 ± 4.7 ^aA^	12.9 ± 1.9 ^aA^
DF-Apple-K30	94.6 ± 21.3 ^A^	0.38 ± 0.06 ^A^	0.40 ± 0.070 ^A^	38.4 ± 15.4 ^A^	14.4 ± 2.5 ^A^
DF-Blackcurrant-K20	171.9 ± 21.4 ^b^	0.46 ± 0.02	0.43 ± 0.063 ^bc^	75.7 ± 18.8 ^b^	35.1 ± 9.5 ^b^
DF-Chokeberry-K20	184.4 ± 25.4 ^b^	0.48 ± 0.03	0.48 ± 0.021 ^c^	88.8 ± 13.6 ^b^	44.9 ± 5.8 ^b^
P-Hibiscus-K20	89.5 ± 24.8 ^a^	0.42 ± 0.02	0.37 ± 0.031 ^b^	33.0 ± 9.052 ^a^	13.8 ± 3.1 ^a^
One-way analysis of variance (ANOVA)					
Factors	*p*-value				
Kale addition (A,B,C)	0.0000 *	0.0013 *	0.0013 *	0.0000 *	0.0000 *
Type of fibre (a,b,c)	0.0002 *	0.1201	0.0000 *	0.0000 *	0.0000 *

*—means significant difference at a confidence level of 0.05; a, b, c and A, B, C—homogeneous groups, the same letters mean no statistically significant differences between the analysed values of indicators; the codes are described in Table 4.

**Table 4 molecules-26-03939-t004:** The recipe of multigrain bars based on the weight of all ingredients, with the addition of kale and fibre preparation or powder of dried hibiscus flower.

Ingredient	Content [%]	Code bars
Whole grain oatmeal (Kupiec, Poland)	20	
Flaxeed (Kresto, Russia)	20	
Sunflower seeds (Bakaland, Poland)	20	
Pumpkin seeds (Bakaland, Poland)	20	
Fresh kale (VitalFresh, Poland)	20 or 30	K20 or K30
Honey (Huzar, Poland)	10	
Water (tap water)	25	
m-40 * apple fibre;	10	DF-Apple
m-40 * chokeberry fibre;	10	DF-Chokeberry
m-40 * blackcurrant fibre	10	DF-Blackcurrant
Dried hibiscus flower—powder (GreenField, Poland)	10	P-Hibiscus

* Coarse fibre, particle size approx. 40 µm.

**Table 5 molecules-26-03939-t005:** Quality features to be assessed and their characteristics.

Sensory Discriminants	Definition	Boundary Terms
Colour	Bar colour (colouring)	10 points—desirable, even 0 points—undesirable, uneven surface colouring
Smell	Intensity of perceived smell	10 points—characteristic for cereal snacks, mild 0 points—imperceptible, atypical
Texture	Fragility and porosity	10 points—desirable, brittle, porous 0 points—undesirable, non-brittle, non-porous, too cohesive
Taste	Taste felt after biting and chewing	10 points—characteristics of cereal snacks 0 points—imperceptible, alien
General suitability	General impression of the quality of the bars	10 points—very desirable 0 points—unacceptable

## Data Availability

Not applicable.

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
