# Peer review of "Development of a High-Fibre Multigrain Bar Technology with the Addition of Curly Kale"

_molecules, 2021, doi:10.3390/molecules26133939_

Round 1

Reviewer 1 Report

This article has confusing structure with the results and conclusions preceding the methods

There is not clear methodology of the addition of kale

Plenty of cereal bars are created every day, there is not anything in particular recipe

The Aw remains high and there is no explanation about the effect to the shelf life. Actually there is no proposed methodology for the shelf life estimation

The DF content and the antioxidant capacity induced by the kale is minimum in relation to the base bar. The base bas is already a good snack, particularly the one with black currants.

What is the point of adding Kale when you have a product that has shorter shelf life, is more expensive due to the addition of Kale or the MA Packaging and has almost no added nutritional value

There is no reference to the EU legislation about nutritional and health claims

Reviewer 2 Report

  1. The abstract is too long, ignore the unnecessary details and focus on aims and main findings.
  2. Some parts of introduction have not been properly cited (e.g. paragrapgh 1), the following references are relevant: a) Nikmaram, N., Garavand, F., Elhamirad, A., Beiraghi-toosi, S., & Goli-Movahhed, G. (2015). Production of high quality expanded corn extrudates containing sesame seed using response surface methodology. Quality Assurance and Safety of Crops & Foods7(5), 713-720. b) Vatankhah, M., Garavand, F., Mohammadi, B., & Elhamirad, A. (2017). Quality attributes of reduced-sugar Iranian traditional sweet bread containing stevioside. Journal of Food Measurement and Characterization11(3), 1233-1239. and c) Ibrahim, S. A., Fidan, H., Aljaloud, S. O., Stankov, S., & Ivanov, G. (2021). Application of Date (Phoenix dactylifera L.) Fruit in the Composition of a Novel Snack Bar. Foods10(5), 918.
  3. Std. deviation should be included in all provided figures and tables.
  4. Figure 2 is not properly labeled, use appropriate graphics to have a better undrestanding
  5. Figure 3, use the same format for the labels (small or capital), aware of this in the whole text (tables and figures)
  6. Double check the inserted letters, for example in Figure 3 (a)
  7. In Figure 4, some error bar labels are missed, correct all carefully.
  8. L348, Figure 1 or 5?? mind the significance letters again
  9. L405: no need to explain the techniques and their importance in this section, only discuss and interpreter the obtained results.
  10. L490, any info about the particle size or the used sieve (mesh) of ground kale?
  11. Proper references should be inserted in the used methodologies
